# Spectral Spatial Traversing in Point Clouds: Enhancing Data Analysis with Mamba Networks

## Abstract

State Space Models (SSMs) such as Mamba have shown significant promise for sequence modeling in Natural Language Processing (NLP) and, more recently, computer vision. This paper presents a new methodology for both supervised and self-supervised learning using Mamba and Masked Autoencoder networks specifically designed for point cloud data. We propose three main contributions that enhance the capability of Mamba networks to process and understand the complex structure of this type of data. The first strategy exploits the spectrum of a graph Laplacian capturing the local connectivity of patches to define an isometry-invariant traversal order of tokens in the Mamba network. Compared to existing point cloud Mamba architectures, which traverse point patches based on a 3D grid, our approach is more robust to the viewpoint and better captures the shape manifold of the point cloud. The second contribution adapts our approach to segmentation using a recursive patch partitioning strategy informed by spectral components of the Laplacian. This strategy enables a more precise integration and analysis point cloud segments. Our last contribution tackles a significant issue in Masked Autoencoder (MAE) for Mamba networks by modifying learnable token placement. Instead of adding them at the end, tokens are restored to their original positions, maintaining essential order and improving learning effectiveness. Extensive experiments confirm our method's superiority over State-Of-The-Art (SOTA) baselines, demonstrating marked improvements in classification, segmentation, and few-shot tasks. The code for this study is available in an *anonymized repository*.

## 1 Introduction

The analysis of 3D point cloud data is fundamental to various applications, including autonomous driving (Qi et al., 2021; Shi et al., 2019), VR/AR (Guo et al., 2020), and robotics (Rusu & Cousins, 2011). Compared to the organized structure of 2D images, point clouds consist of 3D coordinates without direct adjacency information forming an unordered bag. In recent years, considerable efforts have been dedicated to adapt deep learning models such as convolutional neural networks (CNNs) and Transformers to this type of data (Qi et al., 2017b; Yu et al., 2022; Pang et al., 2022; Zhang et al., 2022; Bahri et al., 2024). Due to their permutation invariant self-attention mechanism, Transformer networks are particularly well-suited for the unordered nature of point clouds. However, the quadratic complexity of this mechanism, requiring to compute a weight between each pair of tokens, impedes the application of these networks to large-sized inputs (e.g., 2D images or 3D point clouds represented by many patches). This has prompted researchers to explore more efficient solutions, including the Set Transformer (Lee et al., 2019), Sparse Transformer (Child et al., 2019), Longformer (Beltagy et al., 2020) and Sinkhorn Transformer (Tay et al., 2020).

Recently, methods based on Structured State Space Sequence (S4) (Gu et al., 2021a) such as Mamba (Gu & Dao, 2023) have gained significant traction as a more efficient alternative to Transformers (Liu et al., 2024; Zhu et al., 2024). So far, very few studies have investigated the potential of S4 approaches like Mamba for 3D point clouds. Existing methods like Point-Mamba (Liang et al., 2024) and PCM (Zhang et al., 2024) extend the 2D grid-based traversal employed for images to a 3D grid. However, this straightforward adaptation to point clouds suffers from three crucial problems. **First:** whereas patches from 2D images have adjacency information, which could be exploited by the grid-based traversal, the 3D point patches in point clouds offer a sparse representation of the

object's surface, and nearby patches on a 3D grid are not necessarily adjacent on this surface. **Second:** in the absence of self-attention, task-specific performance is highly influenced by the nature of the token traversal strategy. For example, a traversal suitable for point cloud classification may not be effective for a local task such as point-level classification (i.e., segmentation). **Third:** due to the "direction-sensitive" nature of Mamba, the self-supervised MAE pre-training step of leading point cloud models like Point-MAE (Pang et al., 2022) and Point-M2AE (Zhang et al., 2022) cannot be used directly as there is no attention mechanism to learn the masked tokens' positions.

The contribution of our work focuses on addressing these problems as follows:

1. We introduce a Surface-Aware Spectral Traversing (SAST) strategy based on the Laplacian spectrum of a patch-connectivity graph. Compared to the 3D grid traversal of current approaches like Point-Mamba, our strategy is invariant to isometric transformations (e.g., choice of viewpoint) and better captures the object's surface manifold.

2. We also present a Hierarchical Local Traversing (HLT) for point-level classification (segmentation) that partitions patches recursively based on their spectral coordinates. Unlike our SAST strategy for classification, which considers Laplacian eigenvectors separately in different traversals, this HLT combines them in a single ordering for a more precise modeling of geometry.

3. During the MAE-based Self-Supervised Learning (SSL), we propose a Traverse-Aware Repositioning (TAR) strategy to align the masked tokens according to their spectral adjacency. This strategy addresses the critical issue of spatial adjacency preservation unique to Mamba networks.

## 2 RELATED WORK

**Deep Point Cloud Learning.** With the progress of deep neural networks (DNNs), there has been a growing focus on applying such models to point clouds. Drawing inspiration from models like PointNet (Qi et al., 2017a) and PointNet++ (Qi et al., 2017b), several efforts (Atzmon et al., 2018; Deng et al., 2023; Landrieu & Simonovsky, 2018; Li et al., 2018; Zhao et al., 2019) have been made to develop deep architectures that capture local context information more effectively. Subsequently, models influenced by the Transformer (Vaswani et al., 2017), including versions v1-v3 of the Point Transformer (Wu et al., 2023b; 2022; Zhao et al., 2021) and the Stratified Transformer (Lai et al., 2022), have emerged as leading frameworks, effectively combining local and global data to set new benchmarks. To capitalize on the abundance of unlabeled data, self-supervised pre-training has also emerged as an effective strategy. Notable implementations like Point-BERT (Yu et al., 2022), Point-MAE (Pang et al., 2022), MaskPoint (Liu et al., 2022), Point-M2AE (Zhang et al., 2022), and I2P-MAE (Zhang et al., 2022) have introduced methods for pre-training the Transformer (Vaswani et al., 2017) using techniques based on masked point modeling (Liu et al., 2023; Tang et al., 2023).

Building on the effectiveness of MAE in Text and Image domains, Point-BERT (Yu et al., 2022) presented a revolutionary method inspired by BERT (Devlin et al., 2018), tailoring Transformers to 3D point cloud processing. Point-MAE (Pang et al., 2022) applied MAE-style pre-training to 3D point clouds using a custom Transformer-based Autoencoder (AE) designed to reconstruct masked irregular patches. The use of multi-scale masking and local spatial self-attention mechanisms in Point-M2AE (Zhang et al., 2022) has led to SOTA results in 3D representation learning. Furthermore, I2P-MAE (Zhang et al., 2023) improved self-supervised point cloud processing with a masking strategy leveraging pre-trained 2D models through an Image-to-Point transformation. Point-GPT (Chen et al., 2024) introduced an auto-regressive generative pretraining (GPT) approach to address the unordered nature and low information density of point clouds. Finally, ACT (Dong et al., 2022) proposed a cross-modal knowledge transfer method using pretrained 2D or natural language Transformers as teachers for 3D representation learning.

**State Space Models.** SSMs have long been established in the fields of control theory and signal processing, providing powerful methods for modeling dynamic systems. Drawing from continuous SSMs used in control systems, (Gu et al., 2021b) introduced a Linear State-Space Layer (LSSL) incorporating a continuous-time memorization framework based on the High-Order Polynomial Projection Operator (HiPPO) (Gu et al., 2020) to model long-range dependencies. However, the extensive computational and memory requirements of the state representation make LSSL impractical for standard applications. To address this issue, S4 (Gu et al., 2021a) proposed a method to normalize parameters into a diagonal structure. Subsequently, a variety of structured SSMs have emerged,

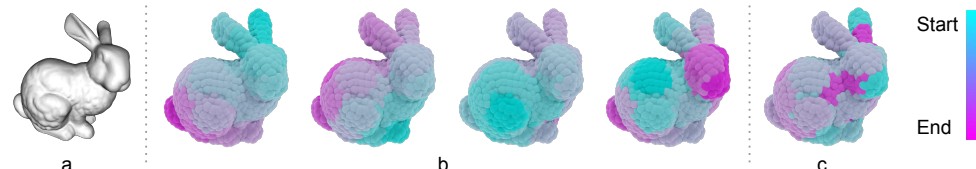

Figure 1: (a) Surface-Aware Spectral Traversing (SAST) over the patched point clouds of a mesh surface. (b) From left to right, traversing based on the first to fourth non-constant smallest eigenvectors. (c) Traversing based on the largest eigenvector forming a fine noncontinuous sequence of tokens.

incorporating complex-diagonal structures (Gupta et al., 2022; Gu et al., 2022), support for multiple-input multiple-output (Gu et al., 2022), and low-rank decomposition (Hasani et al., 2022). These models have subsequently been added into broader representation frameworks (Mehta et al., 2022; Ma et al., 2022b). SGConv (Li et al., 2022) also offers a different method for utilizing S4 as a globally conventional model. To enhance the speed of S4, GSS (Mehta et al., 2022) utilizes a gating structure that decreases the dimensionality of the state space module.

Recently, Mamba (Gu & Dao, 2023) has set a new benchmark by achieving linear-time inference and enhancing the efficiency of the training process. This was accomplished by incorporating selection mechanisms and hardware-aware algorithms into earlier models (Gu et al., 2022; Gupta et al., 2022). MoE-Mamba (Pióro et al., 2024) integrates the Mixture of Experts (MoE) with Mamba, surpassing both standard Mamba and Transformer-MoE models in efficiency.

**SSMs for Vision Tasks** The above-mentioned works primarily focused on the application of SSMs to long-range or causal data types such as language and speech. In the field of vision, a notable study (Liu et al., 2024) proposed the VMamba model which features a Cross-Scan Module (CSM) for enhanced 1D selective scanning in 2D spaces and architectural optimizations that significantly improve its performance and speed across various visual tasks. Another significant paper is Vision Mamba (Zhu et al., 2024) which introduces a novel vision backbone called Vim utilizing bidirectional Mamba blocks.

For point cloud analysis based on Mamba, two key works are Point-Mamba (Liang et al., 2024) and PCM (Zhang et al., 2024). PointMamba introduces a simple approach to token reordering for point cloud analysis by strategically organizing point tokens based on a 3D grid. Similarly, PCM enhances Mamba with a Consistent Traverse Serialization (CTS) technique that converts 3D point clouds into 1D point sequences while maintaining spatial adjacency. Building upon these approaches, our method introduces the Spectral Spatial Traversing (SST) strategy, which improves token ordering and maintains spatial adjacency during MAE-based SSL in Mamba networks.

## 3 METHOD

We begin by outlining the fundamental concepts of SSMs and spectral graph analysis which are at the core of our work. We then give an in-depth presentation of our Surface-Aware Spectral Traversing (SAST) strategy for point cloud processing that improves the model's robustness to isometric transformations and better captures the underlying manifold of the point cloud. Thereafter, we provide detailed specifications of our Hierarchical Local Traversing (HLT) strategy for point-level classification, which defines a more structured patch traversal order based on the recursive partitioning of spectral information. Finally, we introduce our Traverse-Aware Repositioning (TAR), which improves the handling of learnable tokens in masked autoencoders within Mamba networks. Fig. 2 illustrates the overview of the proposed Spectral Spatial Traversing (SST) method.

### 3.1 PRELIMINARIES

**State Space Models (SSMs)** use a series of first-order differential equations to describe how the state of the linear, time-invariant system evolves over time:

$$\dot{h}(t) = Ah(t) + Bx(t), \quad y(t) = Ch(t) + Dx(t), \tag{1}$$

Here, $\dot{h}(t)$ denotes the time derivative of the state vector $h(t)$. The matrices $A$, $B$, $C$, and $D$ are the weighting parameters.

Due to their reliance on continuous data streams $x(t)$, SSMs are not natively equipped to handle discrete inputs represented as $\{x_0, x_1, \ldots\}$. This necessitates the use of a discretized SSM version

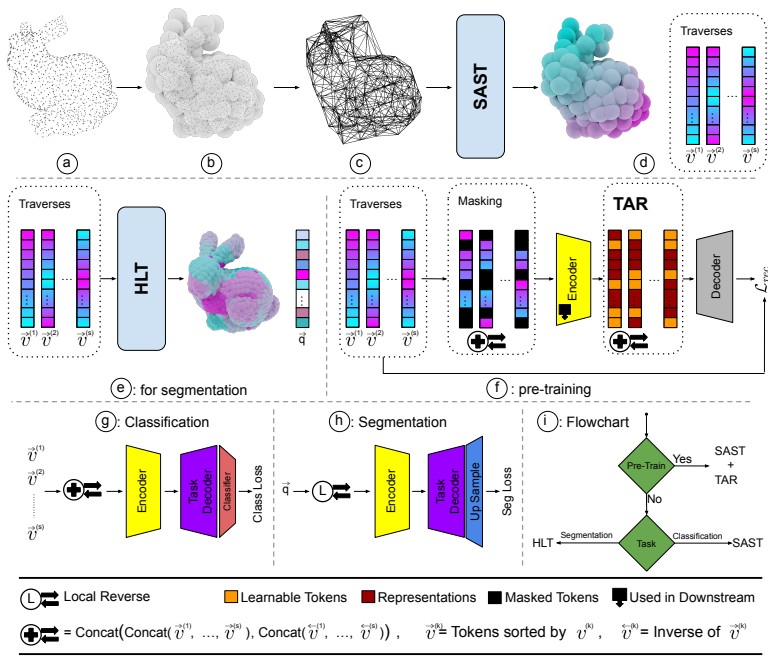

Figure 2: Overview of the proposed Spectral Spatial Traversing (SST) method. (a) Point cloud, (b) Patchification, (c) Forming the adjacency graph, (d) Traversal based on SAST using $s$ smallest eigenvectors, (e) HLT for segmentation tasks, (f) TAR strategy for Masked Autoencoders. The process includes reverse and concatenation operations, with learnable tokens, representations, and masked tokens highlighted. (g) The classification task where tokens are sorted by different eigenvectors first, concatenated, and then fed into the network. (h) The segmentation task where local traversal is applied to q, which is then input into the network. (i) A flowchart visualizing the techniques used in self-supervised learning and various downstream tasks.

for practical applications:

$$h_k = \bar{A}h_{k-1} + \bar{B}x_k, \quad y_k = \bar{C}h_k + \bar{D}x_k. \tag{2}$$

The state space model in its discrete version utilizes a recursive function to link each state $h_k$ to its preceding state, encapsulated by the matrices $\bar{A} \in \mathbb{R}^{N \times N}$, $\bar{B} \in \mathbb{R}^{N \times 1}$, and $\bar{C} \in \mathbb{R}^{N \times 1}$, which are tuned parameter matrices. While matrix $\bar{D} \in \mathbb{R}^{N \times 1}$ may be employed as a residual connection, we follow previous work and exclude it from our model. The transition from a continuous signal representation $x(t)$ to a discrete sequence involves sampling $x(t)$ at intervals defined by $\Delta$, setting each discrete input as $x_k = x(k\Delta)$. This adjustment to a discrete framework results in revised matrix definitions:

$$\bar{A} = (I - \frac{\Delta}{2}A)^{-1}(I + \frac{\Delta}{2}A), \quad \bar{B} = (I - \frac{\Delta}{2}A)^{-1}\Delta B, \quad \bar{C} = C. \tag{3}$$

However, the fixed dynamics of Linear Time-Invariant (LTI) models, exemplified by the constant parameters $A$, $B$, and $C$ in Eq. (3), restrict their capacity to selectively retain or discard relevant information, thereby limiting their contextual awareness. To improve content-aware reasoning, we use Mamba's selection mechanism that manages the propagation and interaction of information across the sequence dimension (Gu & Dao, 2023).

**Spectral Graph Analysis.** Popularized by Chung in the 90s (Chung, 1997), spectral graph analysis characterizes the properties of a graph $G = (V, E)$ by the spectrum (eigenvalues and corresponding eigenvectors) of its Laplacian matrix $L$. This analysis can be understood as a discretized version of the Laplace-Beltrami Operator $\Delta$ of a function $f$ defined on a Riemannian manifold:

$$\Delta f = \text{div}(\text{grad} f) \tag{4}$$

where $\text{grad} f$ is the gradient of $f$ and $\text{div}$ the divergence on the manifold. The solution to the Laplacian eigenvalue problem $\Delta f = -\lambda f$, known as Helmholtz wave equation, is an eigenfunction

corresponding to the natural vibration form of a homogeneous membrane with eigenvalue $\lambda$ (Reuter et al., 2005).

Following methods for spectral clustering (Ng et al., 2001) and normalized cuts (Shi & Malik, 2000), we consider a weighted adjacency matrix $W : V \times V \to \mathbb{R}_+$ where $W_{ij} = 0$ if $(i, j) \notin E$ to model the Euclidean distance of nearby patches (see Section 3.3). The Laplacian matrix of $G$ is defined as $L = D - W$ where $D$ is the diagonal *degree* matrix such that $D_{ii} = \sum_j W_{ij}$. To account for variability in the scale of weights $W_{ij}$ or the distribution of node degrees $D_{ii}$, it is preferable to employ a normalized version of the Laplacian. In this work, we use the Random Walk Laplacian $L_{rw} = I - D^{-1}W$ which has the following useful properties:

1. $L_{rw}$ is positive semi-definite and has $|V|$ non-negative real-valued eigenvalues $0 = \lambda_1 \leq \ldots \leq \lambda_{|V|}$;

2. $0$ is an eigenvalue of $L_{rw}$ with the constant vector as eigenvector and its multiplicity equals the number of connected components in the graph;

3. Following Courant's Nodal Line Theorem (Courant & Hilbert, 2008), the $n$-th eigenmode of $L_{rw}$ has at most $n$ poles of vibration;

4. The representation of a shape by the spectrum of $L_{rw}$ is invariant to isometry (i.e., distance-preserving transformation).

Our method uses the $s$ first non-constant eigenvectors of $L_{rw}$ (i.e., the eigenvectors corresponding to the $s$ smallest non-zero eigenvalues) to define traversal orders for classification (Section 3.3) and segmentation (Section 3.4) that are robust to the viewpoint (due to isometry invariance) and provide a smooth parametrization of the surface manifold. We consider the first eigenvectors as they encode low frequency information (by Courant's Nodal Line Theorem), making the resulting traversal more robust to shape variability and noise. Figure 1 illustrates this concept: (a) shows the original mesh, (b) shows traversals based on the first to fourth non-constant smallest eigenvectors, and (c) shows traversal based on the largest eigenvector forming a non-continuous sequence of tokens.

### 3.2 POINT CLOUD PATCHIFICATION

Given a point cloud $\mathcal{P} = \{p_i\}_{i=1}^{N_p}$, each point represented by 3D coordinates, we convert $\mathcal{P}$ to a reduced set of patches that can be processed more efficiently. Toward this goal, we employ the Farthest Point Sampling (FPS) algorithm to select a subset $\mathcal{C} \subset \mathcal{P}$ of $N_c$ points offering a good coverage of the entire point cloud. These selected points will act as the centers of local patches within the point cloud. For each center point $p_{s_i} \in \mathcal{C}$, we then identify $N_n$ nearest points $\mathcal{N}(p_{s_i}) \subset \mathcal{P}$ using the K-Nearest Neighbours (KNN) algorithm. Following this, each patch is defined as a center $p_{s_i}$ and its corresponding nearest-neighbors $\mathcal{N}(p_{s_i})$.

### 3.3 SURFACE-AWARE SPECTRAL TRAVERSING (SAST)

Current point cloud processing approaches using Mamba, such as Point-Mamba (Liang et al., 2024) and PCM (Zhang et al., 2024), simply extend the 2D grid-based traversal for images to a 3D grid. As mentioned before, this naive strategy suffers from two issues: 1) the 3D grid is view dependent, thus rotating the point cloud or moving the camera yields a different traversal order; 2) unrelated patches may be adjacent in 3D space, hence can be traversed subsequently. To address these problems, we define a traversal order based on the Laplacian spectrum of the patch-connectivity graph.

In this graph, each node corresponds to a patch and the weighted adjacency matrix $W$ is defined using the Euclidean distance between patch centers. For patches $i$ and $j$ defined by center points $p_{s_i}$ and $p_{s_j}$, we add an edge $(i, j)$ if $p_{s_j}$ is among the $K$ nearest neighbors of $p_{s_i}$ or vice-versa. The weight of this edge is computed using a Gaussian kernel: $W_{ij} = \exp\left(-\|p_{s_i} - p_{s_j}\|_2^2/\sigma\right)$ where $\sigma$ is a hyperparameter controlling the kernel width.

Following Section 3.1, we compute the $s$ first non-constant eigenvectors of the Random Walk Laplacian $L_{rw}$. This can be achieved efficiently using an iterative method like the Arnoldi algorithm (Golub & Van Loan, 2013) by exploiting the following facts: 1) matrix $W$ is very sparse, and 2) only the first few eigenvectors need to be computed. Eigenvector $v^{(k)} \in \mathbb{R}^{N_c}$, $k \in \{1, \ldots, s\}$, assigns an eigenfunction value $v_i^{(k)}$ to each patch $i$. In each Mamba block of our model, we perform two separate traversals of tokens (each token corresponds to an input patch) for *every* eigenvector: a forward traversal by increasing value of $v_i^{(k)}$ and a reserve traversal by decreasing value of $v_i^{(k)}$. At

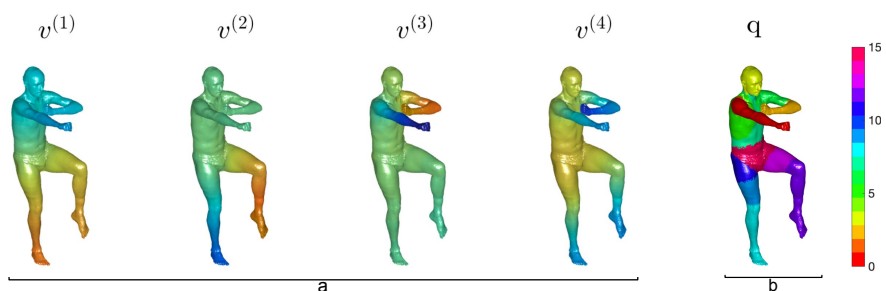

Figure 3: Visualization of the four non-constant smallest Laplacian eigenvectors ($v^{(k)}$, $k = 1, \ldots, 4$) and the discrete partitioning ($q$) of our HLT strategy combining the information of all four eigenvectors. Note: we assumed that patches contain a single point for better visualization.

the end of the block, we concatenate for each token the features computed by the $s \times 2$ traversals. This section is illustrated in Fig. 1 (b), and Fig. 3 (a).

**Canonicalization of spectrum.** Although the spectrum of $L_{rw}$ forms an isometry-invariant representation of the surface manifold, this representation may be impacted by two sources of ambiguity: 1) the sign of eigenvectors is undetermined (i.e., if $v^{(k)}$ is an eigenvector, then so is $-v^{(k)}$), and 2) the order of eigenvectors with similar eigenvalues may vary. We address these two sources of ambiguity with the following canonicalization procedure. For the first one, we flip the sign of an eigenvector $v^{(k)}$ (i.e., $v^{(k)} := -v^{(k)}$) if its first element is negative (i.e., $v_1^{(k)} < 0$). To handle the second ambiguity, we first sort the eigenvectors by non-decreasing eigenvalue. We deal with eigenvalues having a mutliplicity greater than one by finding pairs of consecutive eigenvectors $v^{(k)}$, $v^{(k+1)}$ with near-identical eigenvalues (i.e., $|\lambda_k - \lambda_{k+1}| \leq \epsilon$). For such pairs, we flip the order if $v_1^{(k)} > v_1^{(k+1)}$. This reordering process is repeated until no further change occurs.

### 3.4 HIERARCHICAL LOCAL TRAVERSING (HLT) FOR SEGMENTATION

While effective for classification tasks, the SAST strategy considering each eigenvector in a *separate* traversal may not capture the precise relationship between patches needed for segmentation. To address this issue, we introduce a Hierarchical Local Traversal (HLT) strategy that considers the full spectrum (all $s$ non-constant eigenvectors) simultaneously.

Our strategy is inspired by the recursive binary partitioning technique of normalized cuts (Shi & Malik, 2000). Starting from the canonicalized spectrum (see previous section), tokens are first split based on the first eigenvector $v^{(1)}$, by comparing their corresponding value in $v^{(1)}$ with the mean value $\overline{v}^{(1)}$. This yields a binary partition of tokens $b_i^{(1)} = \mathbb{1}\left(v_i^{(1)} \geq \overline{v}^{(1)}\right) \in \{0, 1\}$ where $\mathbb{1}$ is the indicator function. Each subgroup is then divided based on the mean of the second eigenvector $v^{(2)}$, and so on for other eigenvectors. This partitioning process can be seen as building a binary tree, where each level corresponds to a different eigenvector and leaf nodes $i$ are uniquely identified by the sequence of bits $b_i = [b_i^{(1)}, \ldots, b_i^{(s)}]$ on the path from the root to the leaf. Our HLT method traverses groups of leaf nodes (groups of tokens) sequentially based on the lexicographic order of their binary code (e.g., $[0000], [0001], [0010], [0011], \ldots$ in the case of four eigenvectors). For convenience, we convert binary codes $b_i$ to a non-negative integer $q_i$ (e.g., $bin2Int([0011]) = 3$) and define two traversal orders, by increasing or decreasing values of $q_i$.

For $s$ eigenvectors, the HLT strategy described above divides tokens into $2^s$ segments which are traversed sequentially. In the best case scenario, $\lceil \log_2(N_c) \rceil$ eigenvectors are thus needed to split tokens into individual segments. However, it may happen that multiple tokens fall in the same segment, especially when using fewer eigenvectors. In such case, one can further sort tokens *within* each segment, for example, using the values of the first eigenvector (i.e., $v^{(1)}$). In our implementation, we simply sort these tokens randomly to add stochasticity in the training. This section is illustrated in Fig. 2 (e), and Fig. 3 (b).

As shown in Figure 3, the first Laplacian eigenvectors encode high-level spatial relations (e.g., bottom *vs.* top, left *vs.* right, torso *vs.* limbs, etc.). In the SAST, because these eigenvectors are used in *separate* traversals, the network may not be able to differentiate specific regions/parts of the

point cloud. In contrast, our HLT strategy can capture such specific parts (e.g., head, left or right arm/thigh/calf, pelvis, etc.).

## 3.5 TRAVERSE-AWARE REPOSITIONING (TAR) FOR MASKED AUTOENCODERS

Following state-of-art Transformers for point cloud processing, such as Point-MAE (Pang et al., 2022) and Point-M2AE (Zhang et al., 2022), our method leverages self-supervised pretraining based on MAE to boost performance. In Transformer-based approaches, the learnable tokens of masked patches can be inserted in any position of the sequence (typically at the end) as the self-attention mechanism can still attend to all tokens irrespective of their positions. However, this approach presents a significant problem in Mamba networks which are sensitive to the traversal order of tokens. We handle this problem via a TAR strategy that improves the placement of learnable tokens in MAE within Mamba networks. Specifically, we restore the learnable tokens to their original positions rather than appending them at the end of the sequence. This ensures that the essential order of tokens is maintained, preserving spatial adjacency and enhancing learning effectiveness within Mamba networks.

The proposed TAR strategy selects an arbitrary traversal order and randomly masks a subset of $N_m$ tokens with the same masking ratio as the transformer-based MAEs. These tokens are then removed from the sequence, and their positions are recorded. Afterwards, the remaining (visible) tokens are fed to the encoder that outputs their representation. Before reconstructing the point cloud using the decoder, we reinsert the learnable tokens in the sequence at their recorded position. The same set of masked patches is used for other traversal orders. This procedure can be seen in Fig. 2 (f). Following previous work, we measure the reconstruction error for masked patches using the Chamfer distance:

$$\mathcal{L}_{rec} = \frac{1}{N_m} \sum_{i=1}^{N_m} \text{Chamfer}(\mathcal{S}_i, \hat{\mathcal{S}}_i), \tag{5}$$

where $\mathcal{S}_i \in \mathbb{R}^{N_n \times 3}$ is the set of points forming the $i$-th masked patch and $\hat{\mathcal{S}}_i$ the reconstructed output for these points. The Chamfer distance between two sets of points $\mathcal{S}$ and $\hat{\mathcal{S}}$ is defined as

$$\text{Chamfer}(\mathcal{S}, \mathcal{S}') = \sum_{p \in \mathcal{S}} \min_{p' \in \mathcal{S}'} \|p - p'\|_2^2 + \sum_{p' \in \mathcal{S}'} \min_{p \in \mathcal{S}} \|p - p'\|_2^2. \tag{6}$$

## 4 EXPERIMENTS

Several experiments are conducted to evaluate the proposed method. First, we pretrain the Point-Mamba network using our techniques on the ShapeNet (Chang et al., 2015) training dataset. We then assess the performance of these pretrained models across a variety of standard benchmarks, including object classification, few-shot learning, and segmentation. Additionally, we train the model from scratch on downstream datasets to demonstrate the robustness and versatility of our method. To have a fair comparison, we adopt the masking ratio (60%) that was used in the Point-Mamba model. Moreover, a comprehensive analysis of the computational efficiency, runtime, and memory usage of our SAST approach is provided in the Supplementary Material.

### 4.1 PRETRAINING SETUP

Following Point-Mamba, we adopt the ShapeNet (Chang et al., 2015) dataset for the pretraining and assess the quality of the 3D representations produced by our approach through a linear evaluation on the ModelNet40 (Wu et al., 2015) dataset. The linear evaluation is performed by a Support Vector Machine (SVM) fitted on these features. This classification performance is quantified by the accuracy metric.

### 4.2 DOWNSTREAM TASKS

**Object Classification on Real-World Dataset.** To evaluate our method for point clouds, we test it on the ScanObjectNN dataset (Uy et al., 2019a), as described in previous studies. The augmentation used during training is random rotation. The results, presented in Table 4, show that our strategy significantly improves object classification accuracy in both training from scratch and fine-tuning scenarios. These findings highlight the effectiveness of our approach in enhancing the model's ability to identify and classify objects across various backgrounds, demonstrating its robustness in complex real-world scenarios.

**Object Classification on Clean Objects Dataset.** We also evaluated our method on the ModelNet40 (Wu et al., 2015) dataset, following the protocols established in previous works. The augmentation used during training is scale and transform. As shown in Table 1, our approach achieves notable enhancements on this challenging dataset compared to both the original Point-Mamba and the Transformer-based Point-MAE. This demonstrates the robustness and effectiveness of our method when applied to the Point-Mamba network.

**Few-shot Learning.** We conducted few-shot learning experiments on ModelNet40 (Wu et al., 2015) dataset, adhering to the protocols of previous studies (Pang et al., 2022; Zhang et al., 2022; Liu et al., 2022). The results of our few-shot learning experiments are presented in Table 3. Despite the competitive nature of this benchmark, our method demonstrated outstanding performance across all

Table 1: Object classification on Model-Net40 (Wu et al., 2015). All results are from an input of 1024 points and without voting. Tr: Transformer, and Ma: Mamba networks.

| Methods | Backbone | FLOPs (G) | OA (%) |
|---|---|---|---|
| *Training from scratch* | | | |
| PointNet (Qi et al., 2017a) | - | 0.5 | 89.2 |
| PointNet++ (Qi et al., 2017b) | - | 1.7 | 90.7 |
| PointCNN (Li et al., 2018) | - | - | 92.2 |
| DGCNN (Wang et al., 2019) | - | 2.4 | 92.9 |
| PointNeXt (Qian et al., 2022) | - | 1.6 | 92.9 |
| PCT (Guo et al., 2021) | Tr | 2.3 | 93.2 |
| OctFormer (Wang, 2023) | Tr | - | 92.7 |
| Point-MAE (Pang et al., 2022) | Tr | 2.4 | 92.3 |
| PointMamba (Liang et al., 2024) | Ma | 1.8 | 92.4 |
| Ours | Ma | 1.8 | **92.6** |
| *Training from pretrained* | | | |
| Transformer (Yu et al., 2022) | Tr | - | 92.1 |
| Point-BERT (Yu et al., 2022) | Tr | 2.4 | 92.7 |
| Point-MAE (Pang et al., 2022) | Tr | 2.4 | 93.2 |
| PointMamba (Liang et al., 2024) | Ma | 1.8 | 92.8 |
| Ours | Ma | 1.8 | **93.4** |

Table 2: Part segmentation on the ShapeNet-Part (Yi et al., 2016). The mIoU for all instances (Inst.) is reported. Tr: Transformer, and Ma: Mamba networks. HLT is Hierarchical Local Traversing for segmentation.

| Methods | Backbone | FLOPs (G) | mIoU (%) |
|---|---|---|---|
| *Training from scratch* | | | |
| PointNet (Qi et al., 2017a) | - | - | 83.7 |
| PointNet++ (Qi et al., 2017b) | - | - | 85.1 |
| DGCNN (Wang et al., 2019) | - | - | 85.2 |
| APES (Wu et al., 2023a) | - | - | 85.8 |
| Point-MAE (Pang et al., 2022) | Tr | 15.5 | 85.7 |
| PointMamba (Liang et al., 2024) | Ma | 14.3 | 85.8 |
| Ours (HLT) | Ma | 14.3 | **85.9** |
| *Training from pretrained* | | | |
| Transformer (Yu et al., 2022) | Tr | 15.5 | 85.1 |
| Point-BERT (Yu et al., 2022) | Tr | 15.5 | 85.6 |
| Point-MAE (Pang et al., 2022) | Tr | 15.5 | 86.1 |
| Point-M2AE (Zhang et al., 2022) | Tr | 15.5 | 86.5 |
| Point-GPT-S (Chen et al., 2024) | Tr | - | 86.2 |
| ACT (Dong et al., 2022) | Tr | - | 86.2 |
| I2P-MAE (Zhang et al., 2023) | Tr | - | 86.8 |
| PointMamba (Liang et al., 2024) | Ma | 14.3 | 86.0 |
| Ours (SAST) | Ma | 14.3 | 85.7 |
| Ours (HLT) | Ma | 14.3 | **86.1** |

Table 3: **Few-shot classification on ModelNet40**. We report the average accuracy (%) and standard deviation (%) of 10 independent experiments. '*' denotes reproduced results.

| Method | 5-way | | 10-way | |
|---|---|---|---|---|
| | 10-shot | 20-shot | 10-shot | 20-shot |
| DGCNN (Wang et al., 2019) | 91.8 ±3.7 | 93.4 ±3.2 | 86.3 ±6.2 | 90.9 ±5.1 |
| DGCNN + OcCo (Wang et al., 2021) | 91.9 ±3.3 | 93.9 ±3.1 | 86.4 ±5.4 | 91.3 ±4.6 |
| Transformer (Yu et al., 2022) | 87.8 ±5.2 | 93.3 ±4.3 | 84.6 ±5.5 | 89.4 ±6.3 |
| Transf. + OcCo (Yu et al., 2022) | 94.0 ±3.6 | 95.9 ±2.3 | 89.4 ±5.1 | 92.4 ±4.6 |
| Point-BERT (Yu et al., 2022) | 94.6 ±3.1 | 96.3 ±2.7 | 91.0 ±5.4 | 92.7 ±5.1 |
| Point-M2AE (Zhang et al., 2022) | 96.8 ±1.8 | 98.3 ±1.4 | 92.3 ±4.5 | 95.0 ±3.0 |
| Point-MAE (Pang et al., 2022) | 96.3 ±2.5 | 97.8 ±1.8 | 92.6 ±4.1 | 95.0 ±3.0 |
| PointMamba* (Liang et al., 2024) | 95.9 ±2.1 | 97.3 ±1.9 | 91.6 ±5.3 | 94.5 ±3.5 |
| Ours | **96.4 ±2.7** | **98.5 ±1.5** | **92.0 ±5.1** | **95.1 ±3.6** |

tested scenarios. As shown in Table 3, our Mamba-based method achieves results comparable to or exceeding those of transformer-based methods (Point-MAE and Point-M2AE).

**Part Segmentation.** Our method's capacity for representation learning was assessed using the ShapeNetPart dataset (Yi et al., 2016), following the same experimental settings as in prior studies (Qi et al., 2017a;b; Yu et al., 2022). Table 2 presents the results of various methods on a highly challenging dataset. As observed, the trend of improvement in previous methods is minor, indicating the difficulty of achieving significant performance gains on this dataset. For instance, Point-GPT (Chen et al., 2024) and ACT (Dong et al., 2022), despite being state-of-the-art and complex methods, show only marginal improvements over each other and other state-of-the-art approaches. Similarly, I2P-MAE (Zhang et al., 2023), which supplements 3D data with additional 2D information and uses the Point-M2AE backbone, also fails to achieve significant improvements in segmentation compared to Point-M2AE. For the "Training from scratch" setting, our method outperforms several state-of-the-art approaches. In the "Training from pretrained" setting, we further demonstrate the effectiveness of HLT strategy compared to SAST in the segmentation task. Given the difficulty of achieving major gains in this domain, our results are reasonable and follow the observed trend.

## 4.3 ABLATION STUDIES

In this section, we aim to investigate the effects of different parameters on our method. We will focus on two key aspects: the effect of the number of non-constant smallest eigenvectors and the adjacency matrix used in the SAST strategy, and the analysis of the TAR strategy.

**Analysis of Eigenvectors and Graph.** One of our ablation studies investigates the impact of the number of non-constant smallest eigenvectors used in the SAST and TAR strategies. As depicted in Fig. 4 (**left**), the best performance is achieved when using the four non-constant smallest eigenvectors (blue line) for traversing. When the number of non-constant smallest eigenvectors increases beyond four, the performance drops. This is because the additional eigenvectors are closer to the largest eigenvectors, which are less smooth and do not capture the most significant structural variations effectively. For comparison, the green line represents the performance of the Point-Mamba (Liang et al., 2024) model, and the orange line represents the Point-MAE (Pang et al., 2022) model. Finally, the brown line indicates the performance of the Point-Mamba model without any traversing, capturing

Table 4: Object classification on ScanObjectNN (Uy et al., 2019b). Accuracy (%) is reported. [†] indicates that this method was fine-tuned without rotation augmentation.

| Methods | Backbone | Param. (M) | FLOPs (G) | OBJ-BG | OBJ-ONLY | PB-T50-RS |
|---------|----------|------------|-----------|--------|----------|-----------|
| *Training from scratch* | | | | | | |
| PointNet (Qi et al., 2017a) | - | 3.5 | 0.5 | 73.3 | 79.2 | 68.0 |
| PointNet++ (Qi et al., 2017b) | - | 1.5 | 1.7 | 82.3 | 84.3 | 77.9 |
| DGCNN (Wang et al., 2019) | - | 1.8 | 2.4 | 82.8 | 86.2 | 78.1 |
| PRANet (Cheng et al., 2021) | - | - | - | - | - | 81.0 |
| PointNeXt (Qian et al., 2022) | - | 1.4 | 1.6 | - | - | 87.7 |
| PointMLP (Ma et al., 2022a) | - | 13.2 | 31.4 | - | - | 85.4 |
| RepSurf-U (Ran et al., 2022) | - | 1.5 | 0.8 | - | - | 84.3 |
| ADS (Hong et al., 2023) | - | - | - | - | - | 87.5 |
| Transformer (Yu et al., 2022) | Transformer | 22.1 | 4.8 | 79.86 | 80.55 | 77.24 |
| Point-MAE (Pang et al., 2022) | Transformer | 22.1 | 4.8 | 86.75 | 86.92 | 80.78 |
| PointMamba(Liang et al., 2024) | Mamba | 12.3 | 3.6 | 90.87 | 90.18 | 85.60 |
| Ours | Mamba | 12.3 | 3.6 | **92.42** | **91.39** | **87.61** |
| *Training from pretrained* | | | | | | |
| Transformer (Yu et al., 2022) | Transformer | 22.1 | 4.8 | 79.86 | 80.55 | 77.24 |
| Transformer-OcCo (Yu et al., 2022) | Transformer | - | - | 84.85 | 85.54 | 78.79 |
| Point-BERT (Yu et al., 2022) | Transformer | 22.1 | 4.8 | 87.43 | 88.12 | 83.07 |
| Point-M2AE[†] (Zhang et al., 2022) | Transformer | - | - | 91.22 | 88.81 | 86.43 |
| Point-MAE (Pang et al., 2022) | Transformer | 22.1 | 4.8 | 92.77 | 91.22 | 89.04 |
| PointMamba (Liang et al., 2024) | Mamba | 12.3 | 3.6 | 93.29 | 91.56 | 88.17 |
| PCM (Zhang et al., 2024) | Mamba | 12.3 | 3.6 | - | - | 86.9 |
| Ours | Mamba | 12.3 | 3.6 | **94.32** | **92.08** | **89.10** |

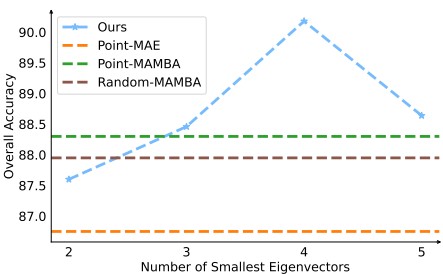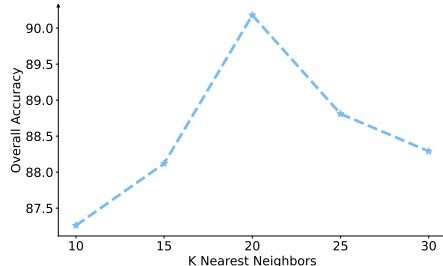

Figure 4: Analysis of the number of non-constant smallest eigenvectors and comparison with previous methods (**left**) and Analysis of the number of nearest neighbors $K$ (**right**).

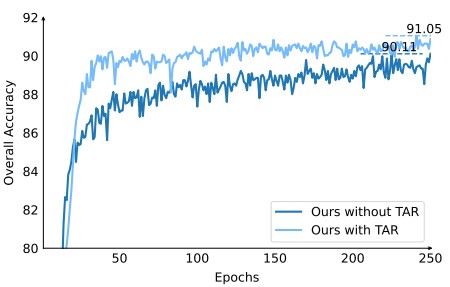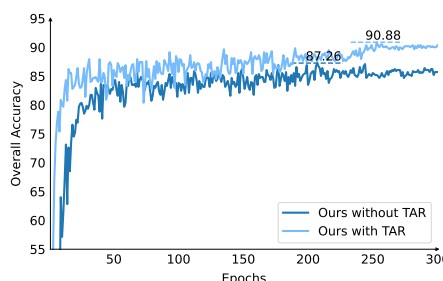

Figure 5: The effect of the TAR strategy in the pretraining phase (left) and in finetuning (right).

input without specific ordering. The significantly lower performance of this model compared to ours underscores the importance of appropriate traversing for Mamba networks. All of these methods are trained and tested on the ScanObjectNN dataset (Uy et al., 2019a) (OBJ-BG) from scratch with Scale and Transform augmentation.

Another study examines the impact of the number of nearest neighbors used in creating the adjacency matrix. As shown in Fig. 4 (**right**), the best performance (blue line) is achieved with 20 nearest neighbors. The model's accuracy increases as the number of nearest neighbors increases from 10 to 20, reaching its peak at 20 nearest neighbors. Beyond this point, the performance starts to decline.

**Analysis of TAR Strategy.** One of the studies examines the impact of the TAR strategy on the model's performance. As shown in Fig. 5 (**left**), the accuracy of the model with and without the TAR strategy is plotted against the number of epochs. The model incorporating the TAR strategy (light blue line) demonstrates superior performance compared to the model without TAR (dark blue line). This figure relates to the pretraining phase on the ShapeNet (Chang et al., 2015) dataset, which is subsequently tested on the ModelNet (Wu et al., 2015) dataset using a SVM. The final accuracy achieved with the TAR strategy is 91.05%, whereas the model without TAR achieves a lower accuracy of 90.11%. This improvement highlights the significance of the TAR strategy, which restores the learnable tokens to their original positions rather than appending them at the end of the sequence to maintain spatial adjacency and positional information during the training process.

Additionally, we show the effect of the TAR strategy in a downstream task. Pretrained models with and without the TAR strategy are fine-tuned on the ScanObjectNN dataset (Uy et al., 2019a) (OBJ-BG). As shown in Fig. 5 (**right**), the model pretrained with the TAR strategy (light blue line) achieves significantly higher overall accuracy compared to the model without the TAR strategy (dark blue line). This demonstrates that the model pretrained with the TAR strategy has learned more meaningful features, leading to better performance in the downstream task.

## 5 CONCLUSION

We introduced three strategies to enhance Mamba networks for point cloud data: isometry-invariant token traversal, recursive patch partitioning for segmentation, and improved learnable token placement. Our methods demonstrate superior performance over state-of-the-art baselines in classification, segmentation, and few-shot tasks.

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
