# OpenReview forum: "Spectral Spatial Traversing in Point Clouds: Enhancing Data Analysis with Mamba Networks"
_ICLR.cc/2025/Conference — ICLR 2025 Conference Withdrawn Submission_

### Official Review · Reviewer_bw18 · 2024-10-31

**Soundness:** 2
**Presentation:** 3
**Contribution:** 3
**Rating:** 5
**Confidence:** 4

**Summary:**

The paper presents a new design for applying Mamba state-space models onto point clouds, with the focus on traversing the points in an order that is derived from spectral analysis of the 3D shapes. In particular, the paper proposes to find the low-frequency eigen modes of a shape by solving the eigen vectors of the graph Laplacian on the point cloud, and traverse the points according to their ordering in the eigen modes. The point traversal thus defined is invariant to rotations and other isometries of the 3D point cloud, and points close with respect to surface distance are grouped together, therefore providing more meaningful sequences for Mamba models to learn effective representations that are useful for downstream tasks like classification and segmentation. The paper also proposes a pretraining scheme based on masked autoencoding, where the tokens are always ordered as mentioned before, to improve representation learning.

While the results are impressive, I expect a sound discussion of the difference between SAST (i.e. separate ordering by different eigen vectors) and HLT (i.e. a single lexicographical ordering by combining the different eigen vectors), to better appreciate their applicability across tasks.

**Strengths:**

The proposed point traversal is derived from the intrinsic spectral analysis of a 3D shape. It is therefore invariant to extrinsic transforms like rigid transforms and other isometric deformations. The paper shows such an ordering enhances the learning of geometric features, and fits nicely to mamba sequential models.

The paper applies the technique to different tasks for point cloud deep learning, including classification, segmentation and pretraining. For the global classification and local segmentation tasks, the paper proposes appropriate modifications of sequence construction, where the classification uses ordering by different eigenvectors separately, and segmentation uses a lexicographical combination of eigenvectors to provide per-point spectral encoding. For pretraining via MAE, the masked tokens are positioned inline the ordering.

Experiments have shown the new mamba point cloud network improves over baselines, including point transformers for both computational cost and accuracy, and previous mamba networks based on extrinsic grid ordering.

**Weaknesses:**

The review of state-space models (up to Eq(3)) does not include Mamba specific details, with its essential selection mechanism skipped.

Fig. 2 can be enlarged to enhance readability. "Local reverse" is not defined, either in the figure or in the main text.

The network structures for encoders and decoders, as well as classification and segmentation heads, are not given and not clear. The traverse-aware repositioning strategy can be illustrated in more details. In particular, what are the input tokens representing masked tokens in the decoder? Are they learnable? Do they contain positional embedding information?

The paper does not discuss limitations. There are no tests on scene level point clouds. Do the authors expect challenges with hyperparameter tuning for a different set of data like scenes?

**Questions:**

In addition to the questions above, can the authors provide any insight of why such a mamba network can beat transformer based networks, which are computationally more expensive but also more comprehensive by being able to model pairwise interactions among all points. In comparison for LLMs, mamba networks mainly improve computational efficiency but not accuracy over transformers.

In the appendix, it's shown that the network is computationally more efficient than point-mamba. What are the reasons for this difference, when both use mamba networks?

In the appendix, the Hierarchical Local Traversing scheme is applied to classification and produces poor results; the HLT scheme for classification is not as competent as point-mamba. Any insight why it's worse than the separate ordering by different eigenvectors? Does the separate traversing provide a natural multiscale analysis that is essential for global classification? To test this hypothesis, HLT truncated at different eigenvalues can be combined to mimic the multiscale analysis, and one can see if such a multiscale HLT can improve over the single scale HLT for classification and segmentation.

On the other hand, what if SAST is applied for segmentation? Is there a particular difficulty in formulating segmentation based on SAST ordering by separate eigenvectors?

---

### Official Review · Reviewer_6tEF · 2024-10-31

**Soundness:** 2
**Presentation:** 2
**Contribution:** 2
**Rating:** 3
**Confidence:** 5

**Summary:**

This work introduces a novel approach for point cloud processing with three key strategies: (1) a Surface-Aware Spectral Traversing (SAST) method based on Laplacian spectrum analysis, enhancing isometric transformation invariance and surface manifold capture compared to traditional 3D grid traversal; (2) a Hierarchical Local Traversing (HLT) strategy for segmentation, using recursive spectral partitioning for precise geometry modeling; and (3) a Traverse-Aware Repositioning (TAR) strategy within MAE-based Self-Supervised Learning (SSL), aligning masked tokens to maintain spectral adjacency and address spatial adjacency preservation in Mamba networks.

**Strengths:**

1. This paper proposes a new approach to enable Mamba to process point clouds
2. This paper introduces MAE to train SSM for point clouds

**Weaknesses:**

1. While the authors propose Mamba as a more efficient approach for point clouds, the evaluation is limited and does not fully substantiate this claim. Showing the results requested in Question 1 would better support the claim.
2. In Section 3.3, the authors hypothesize potential issues with PointMamba[1] or PCM[2] (Lines 254-257), yet provide minimal experimental evidence to support it.
3. The claimed efficiency of Mamba over Transformers lacks sufficient testing on data with long input sequences, which would better demonstrate Mamba’s advantages.

**Questions:**

1. Aside from the FLOPs comparison, to better evaluate the efficiency of your method, could you provide a runtime or FPS analysis between your method and a transformer-based approach like Point-MAE?
2. Could you offer additional experiments to address Weakness #2? For instance, by applying SAST to PointMamba [1] or PCM [2], or integrating their strategies into your model.
3. Would it be feasible to use scene-level point clouds or increase point sampling from object meshes to address the issue raised in Weakness #3?

**References**

[1] Dingkang Liang, Xin Zhou, Xinyu Wang, Xingkui Zhu, Wei Xu, Zhikang Zou, Xiaoqing Ye, and Xiang Bai. Pointmamba: A simple state space model for point cloud analysis.

[2] Tao Zhang, Xiangtai Li, Haobo Yuan, Shunping Ji, and Shuicheng Yan. Point could mamba: Point cloud learning via state space model.

---

### Official Review · Reviewer_xyDr · 2024-11-02

**Soundness:** 3
**Presentation:** 3
**Contribution:** 3
**Rating:** 6
**Confidence:** 3

**Summary:**

his paper presents a novel methodology for both supervised and self-supervised learning using Mamba and Masked Autoencoder networks specifically designed for point cloud data. The authors propose three significant contributions aimed at enhancing the capability of Mamba networks in processing and understanding the complex structures inherent in this data type.

Firstly, the authors exploit the spectrum of a graph Laplacian to capture the local connectivity of patches, thereby defining an isometry-invariant traversal order of tokens in the Mamba network. This approach demonstrates greater robustness to viewpoint variations compared to existing point cloud Mamba architectures that utilize a 3D grid traversal, effectively capturing the shape manifold of the point cloud.

Secondly, the methodology is adapted for segmentation through a recursive patch partitioning strategy informed by the spectral components of the Laplacian. This strategy facilitates a more precise integration and analysis of point cloud segments.

Lastly, the authors address a notable issue in the Masked Autoencoder (MAE) for Mamba networks by modifying the placement of learnable tokens. Instead of positioning these tokens at the end, the authors restore them to their original locations, thereby maintaining essential order and enhancing learning effectiveness.

Extensive experiments are conducted to validate the proposed method, revealing its superiority over State-Of-The-Art (SOTA) baselines with marked improvements in classification, segmentation, and few-shot tasks.

**Strengths:**

- The SAST traversal method ensures isometry-invariance, preserving surface features under rotation and viewpoint changes.
- The HLT approach offers fine-grained segmentation by recursively partitioning based on spectral coordinates.
- The TAR strategy repositions tokens to retain spatial adjacency, enhancing learning without needing an attention mechanism.
- The paper is thoroughly detailed, with informative figures that significantly aid in understanding the concepts.
- The experimental section provides a strong validation of the proposed methods through diverse tasks.
- Code is provided, ensuring reproducibility and enabling further research.

**Weaknesses:**

1. In an HLT segmentation strategy, how sensitive might the segmentation accuracy be to the number of feature vectors used? Does using fewer or more feature vectors affect the partition boundary?
2. The HLT strategy provides detailed segmentation through multi-level recursive partitioning. However, could segmentation accuracy be impacted on point clouds with noisy boundaries or high noise levels?
3. Since the TAR strategy involves reordering tokens, does the change in traversal order have much impact on the model?

**Questions:**

1. In an HLT segmentation strategy, how sensitive might the segmentation accuracy be to the number of feature vectors used? Does using fewer or more feature vectors affect the partition boundary?
2. The HLT strategy provides detailed segmentation through multi-level recursive partitioning. However, could segmentation accuracy be impacted on point clouds with noisy boundaries or high noise levels?
3. Since the TAR strategy involves reordering tokens, does the change in traversal order have much impact on the model?

---

### Official Review · Reviewer_NxeQ · 2024-11-03

**Soundness:** 3
**Presentation:** 2
**Contribution:** 2
**Rating:** 5
**Confidence:** 3

**Summary:**

This paper proposes a novel graph Laplacian-based serialization method for point cloud data, aimed at improving point cloud segmentation tasks in models like Mamba. The method first sorts the point cloud using a recursive patch partitioning strategy informed by spectral components of the Laplacian, capturing both geometric and semantic information. To ensure that sorted patches retain consistent ordering, the method utilizes a token restoration strategy, allowing patches to be restored to their original positions. Additionally, the authors extend the Mamba model to incorporate Masked Autoencoder (MAE) pretraining for point clouds. Experimental results demonstrate that the proposed approach outperforms baseline models provided in the paper, but it does not yet surpass state-of-the-art results achieved by models such as Mamba3D [1]. The paper contributes an effective integration of MAE pretraining for enhanced point cloud segmentation performance.

[1] Han, Xu, et al. "Mamba3d: Enhancing local features for 3d point cloud analysis via state space model." ACM MM (2024).

**Strengths:**

The paper presents an original approach to point cloud processing by introducing a graph Laplacian-based serialization method, which leverages spectral properties for ordering point cloud patches. The recursive patch partitioning strategy based on Laplacian eigenvectors provides a fresh perspective on capturing both local and global structures within point clouds. Furthermore, the integration of Masked Autoencoder (MAE) pretraining with the Mamba model for 3D segmentation represents a creative extension, leveraging recent advances in self-supervised learning and adapting them to the 3D domain.

**Weaknesses:**

The paper claims to adapt the proposed method to segmentation tasks using a recursive patch partitioning strategy informed by spectral components of the Laplacian, suggesting broader applicability to various segmentation tasks. However, the experiments are restricted to part segmentation, which limits the evaluation of the method’s effectiveness on other segmentation tasks, particularly scene segmentation. Scene segmentation tasks on datasets like ScanNet and S3DIS would be valuable additions to demonstrate the method’s robustness and adaptability to complex real-world environments with diverse objects, occlusions, and varying scales.
While the paper emphasizes the importance of the Hierarchical Local Traversal (HLT) strategy, the ablation study does not provide a detailed analysis of its effectiveness.
Currently, the lack of visualizations limits the interpretability of the results and the ability to understand how well the model captures. Visualizations of part segmentation results and feature embeddings would allow readers to gain insights into the quality of the model’s learned representations and the effectiveness of the MAE pretraining.

**Questions:**

[1] The method claims to be adaptable for segmentation tasks but only reports results on part segmentation. Have you considered testing the method on scene segmentation tasks, such as ScanNet or S3DIS? If not, could you elaborate on potential challenges or limitations that might arise when applying your approach to scene-level datasets?

[2] The Hierarchical Local Traversal (HLT) strategy is presented as a core contribution. Could you provide an ablation study to isolate its effectiveness? Specifically, how does the segmentation performance change when using simpler patching strategies ?

[3] The proposed method demonstrates improvements over baseline models, but it does not achieve state-of-the-art performance compared to models like Mamba3D. Could you discuss potential reasons for this performance gap and any future improvements that might help bridge it?

[4] The paper lacks visualizations of the part segmentation results and pre-trained feature space, which could help readers assess the model’s qualitative performance. Could you provide visual comparisons of segmented point clouds and/or feature embeddings (e.g., using t-SNE or UMAP) for different parts or classes?

---

### Note · Authors · 2024-11-14

I have read and agree with the venue's withdrawal policy on behalf of myself and my co-authors.